

# Non-synonymous to synonymous substitutions suggest that orthologs tend to keep their functions, while paralogs are a source of functional novelty

Juan M. Escorcia-Rodríguez[1], Mario Esposito[2],
Julio A. Freyre-González[1] and Gabriel Moreno-Hagelsieb[2]

[1] Regulatory Systems Biology Research Group, Program of Systems Biology, Center for Genomic Sciences, Universidad Nacional Autonóma de México, Cuernavaca, Morelos, México
[2] Department of Biology, Wilfrid Laurier University, Waterloo, Canada

## ABSTRACT

Orthologs separate after lineages split from each other and paralogs after gene duplications. Thus, orthologs are expected to remain more functionally coherent across lineages, while paralogs have been proposed as a source of new functions. Because protein functional divergence follows from non-synonymous substitutions, we performed an analysis based on the ratio of non-synonymous to synonymous substitutions (dN/dS), as proxy for functional divergence. We used five working definitions of orthology, including reciprocal best hits (RBH), among other definitions based on network analyses and clustering. The results showed that orthologs, by all definitions tested, had values of dN/dS noticeably lower than those of paralogs, suggesting that orthologs generally tend to be more functionally stable than paralogs. The differences in dN/dS ratios remained suggesting the functional stability of orthologs after eliminating gene comparisons with potential problems, such as genes with high codon usage biases, low coverage of either of the aligned sequences, or sequences with very high similarities. Separation by percent identity of the encoded proteins showed that the differences between the dN/dS ratios of orthologs and paralogs were more evident at high sequence identity, less so as identity dropped. The last results suggest that the differences between dN/dS ratios were partially related to differences in protein identity. However, they also suggested that paralogs undergo functional divergence relatively early after duplication. Our analyses indicate that choosing orthologs as probably functionally coherent remains the right approach in comparative genomics.

# INTRODUCTION

Since the beginning of comparative genomics, the assumption was made that orthologs could be expected to conserve their functions more often than paralogs (*Mushegian & Koonin, 1996*; *Huynen & Bork, 1998*; *Bork et al., 1998*; *Tatusov et al., 2000*). The expectation is based on the definitions of each homolog type: orthologs are characters

Corresponding author
Gabriel Moreno-Hagelsieb, gmoreno@wlu.ca

separating after speciation events, while paralogs are characters separating after duplication events (*Fitch, 2000*). Given those definitions, orthologs could be considered the "same" genes in different species, while paralogy has been proposed as a mechanism for the evolution of new functions, under the argument, in very simplified terms, that one of the copies could maintain the original function, while the other copy would have some freedom to functionally change (*Ohno, 1970*). This neither means that orthologs cannot evolve new functions, nor that paralogs necessarily evolve new functions. However, a scenario whereby most orthologs would diverge in functions at a higher rate than paralogs seems far from parsimonious, thus very unlikely. Therefore, it has been customary to use some working definition of orthology to infer the genes whose products most likely perform the same functions across different lineages (*Mushegian & Koonin, 1996*; *Huynen & Bork, 1998*; *Bork et al., 1998*; *Tatusov et al., 2000*; *Gabaldón & Koonin, 2013*).

Despite such a straightforward expectation, a report was published making the surprising claim that orthologs diverged in function more often than paralogs (*Nehrt et al., 2011*). The controversial article was mainly based on the comparison of Gene Ontology annotations among orthologs and paralogs from two species: humans and mice (*Nehrt et al., 2011*). If the report were correct, it would mean, for example, that mice myoglobin could be performing the function that human alpha-haemoglobin performs. However, data in the article showed that paralogs found within a genome, had more consistent gene ontology annotations than any homologs between both genomes. This was true even for identical proteins. Thus, rather than functional differences, it was possible that annotations of homologous genes were more consistent within a genome than between genomes. Accordingly, later work showed that gene ontologies suffered from "ascertainment bias", which made annotations more consistent within an organism than without (*Thomas et al., 2012*; *Altenhoff et al., 2012*). Later work showed gene expression data suggesting that orthologs had more coherent functions than paralogs (*Kryuchkova-Mostacci & Robinson-Rechavi, 2016*).

We thus wondered whether we could perform some analyses that did not suffer from annotation bias, and that could cover most of the homologs found between any pair of genomes, even if they had no functional annotations. Given that changes in protein function require changes in amino acids, analyses of non-synonymous to synonymous substitution rates, which compare the relative rates of positive and negative (purifying) selection (*Ohta, 1995*; *Yang & Nielsen, 2000*), might serve as proxies for functional divergence. The most functionally stable homologs would be expected to have lower $dN/dS$ ratios compared to less functionally stable homologs. Thus, comparisons between the $dN/dS$ distributions of orthologs and paralogs could show differences in their tendencies to conserve their functions. Since most of the related works have focused on eukaryotes, we centered our analyzes on prokaryotes (Bacteria and Archaea). We used five working definitions of orthology, including RBH, which is the foundation of most graph-based orthology prediction methods, besides arguably being the most usual working definition of orthology (*Altenhoff & Dessimoz, 2009*; *Wolf & Koonin, 2012*; *Galperin et al., 2019*).

**Table 1 Genomes used in this study.**

| Genome ID | Class | Order | Species |
|---|---|---|---|
| Phylum proteobacteria | | | |
| GCF_000005845 | Gammaproteobacteria | Enterobacterales | *Escherichia coli* |
| GCF_002370525 | Gammaproteobacteria | Pseudomonadales | *Acinetobacter guillouiae* |
| GCF_002847445 | Alphaproteobacteria | Rhodobacterales | *Paracoccus zhejiangensis* |
| GCF_004194535 | Betaproteobacteria | Neisseriales | *Iodobacter fluviatilis* |
| GCF_013085545 | Deltaproteobacteria | Desulfovibrionales | *Desulfovibrio marinus* |
| GCF_013283835 | Epsilonproteobacteria | Campylobacterales | *Poseidonibacter lekithochrous* |
| GCF_000317895 | Oligoflexia | Bdellovibrionales | *Bdellovibrio bacteriovorus* |
| GCF_009662475 | Acidithiobacillia | Acidithiobacillales | *Acidithiobacillus thiooxidans* |
| GCF_002795805 | Zetaproteobacteria | Mariprofundales | *Mariprofundus aestuarium* |
| GCF_003574215 | Hydrogenophilalia | Hydrogenophilales | *Hydrogenophilus thermoluteolus* |
| Phylum firmcutes | | | |
| GCF_000009045 | Bacilli | Bacillales | *Bacillus subtilis* |
| GCF_002197645 | Bacilli | Lactobacillales | *Enterococcus wangshanyuanii* |
| GCF_000218855 | Clostridia | Eubacteriales | *Clostridium acetobutylicum* |
| GCF_003991135 | Clostridia | Halanaerobiales | *Anoxybacter fermentans* |
| GCF_000020005 | Clostridia | Natranaerobiales | *Natranaerobius thermophilus* |
| GCF_003966895 | Negativicutes | Selenomonadales | *Methylomusa anaerophila* |
| GCF_003367905 | Negativicutes | Veillonellales | *Megasphaera stantonii* |
| GCF_012317185 | Erysipelotrichia | Erysipelotrichales | *Erysipelatoclostridium innocuum* |
| GCF_000299355 | Tissierellia | Tissierellales | *Gottschalkia acidurici* |
| GCF_001544015 | Limnochordia | Limnochordales | *Limnochorda pilosa* |
| Phylum euryarchaeota | | | |
| GCF_000025625 | Halobacteria | Natrialbales | *Natrialba magadii* |
| GCF_000011085 | Halobacteria | Halobacteriales | *Haloarcula marismortui* |
| GCF_000025685 | Halobacteria | Haloferacales | *Haloferax volcanii* |
| GCF_000195895 | Methanomicrobia | Methanosarcinales | *Methanosarcina barkeri* |
| GCF_000013445 | Methanomicrobia | Methanomicrobiales | *Methanospirillum hungatei* |
| GCF_001433455 | Thermococci | Thermococcales | *Thermococcus barophilus* |
| GCF_000024185 | Methanobacteria | Methanobacteriales | *Methanobrevibacter ruminantium* |
| GCF_000006175 | Methanococci | Methanococcales | *Methanococcus voltae* |
| GCF_000734035 | Archaeoglobi | Archaeoglobales | *Archaeoglobus fulgidus* |
| GCF_000007185 | Methanopyri | Methanopyrales | *Methanopyrus kandleri* |

Note:
The query genomes were the first in each group.

## MATERIALS AND METHODS

### Genome data

We downloaded the analyzed genomes from NCBI's RefSeq Genome database (*Haft et al., 2018*). We performed our analyses by selecting genomes from three taxonomic phyla, using one genome within each phylum as a query genome (Table 1): *Escherichia coli* K12 MG1655 (phylum Proteobacteria, domain Bacteria, assembly ID: GCF_000005845),

*Bacillus subtilis* 168 (Firmicutes, Bacteria, GCF_000009045), and *Natrialba magadii* ATCC43099 (Euryarchaeota, Archaea, GCF_000025625).

## Orthologs

We used five working definitions of orthology:

### Reciprocal best hits (RBH)

We compared the proteomes of each of these genomes against those of other members of their taxonomic phylum using diamond (*Buchfink, Xie & Huson, 2015*), with the $--very-sensitive$ option, and a maximum e-value of $1 \times 10^{-6}$ (-evalue 1e−6) (*Hernández-Salmerón & Moreno-Hagelsieb, 2020*). We also required a minimum alignment coverage of 60% of the shortest sequence. Orthologs were defined as reciprocal best hits (RBH) as described previously (*Moreno-Hagelsieb & Latimer, 2008*; *Ward & Moreno-Hagelsieb, 2014*; *Hernández-Salmerón & Moreno-Hagelsieb, 2020*). Except where noted, paralogs were all matches left after finding RBH.

### Ortholog groups with inparalogs (InParanoid)

InParanoid is a graph-based tool to identify orthologs and in-paralogs from pairwise sequence comparisons (*Sonnhammer & Östlund, 2015*). InParanoid first runs all-*vs*-all blastp and identifies RBH. Then, it uses the RBH as seeds to identify co-orthologs for each gene (which the authors define as in-paralogs), proteins from the same organism that obtain better bits score than the RBH. Finally, through a series of rules, InParanoid cluster the co-orthologs to return non-overlapping groups. The authors define outparalogs as those blast-hits outside of the co-ortholog clusters (*Sonnhammer & Östlund, 2015*).

We ran InParanoid for each query genome against those of other members of their taxonomic order. InParanoid was run with the following parameters: double blast and 40 bits as score cutoff. The first pass run with compositional adjustment on and soft masking. This removes low complexity matches but truncates alignments (*Sonnhammer & Östlund, 2015*). The second pass run with compositional adjustment off to get full-length alignments. We used as in-paralogs the combinatorial of the genes of the same organism from the same cluster, and as out-paralogs those blast-hits outside of the co-ortholog clusters.

### Orthologous MAtrix (OMA)

OMA is a pipeline and database that provides three different types of orthologs: pairwise orthologs, OMA groups (orthogroups), and hierarchical orthologous groups (*Zahn-Zabal, Dessimoz & Glover, 2020*). OMA makes an effort to remove xenologs by using a third proteome as witness of non-orthology (*Roth, Gonnet & Dessimoz, 2008*). To the best of our knowledge, OMA is the only orthology prediction method, still being maintained, able to deal with xenology. The OMA pipeline for the identification of orthologs is based on best reciprocal Smith-Waterman hits and some tolerance for evolutionary distance that allows for co-orthology. For pairwise orthology identification, a verification step to detect xenologs is applied using a third proteome that retained both pseudo-orthologous genes (*Train et al., 2017*).

We ran the OMA standalone (version 2.5.0) with all the proteomes for each taxonomic group using the default parameters (*Train et al., 2017*). We used the pairwise orthology outputs considering the query organisms. For the identification of in-paralogs for the query organism, we used the co-orthologous genes mapping to one or more orthologs in the rest of the organisms. OMA also generates pairwise paralogy outputs, including former candidates for orthologs that did not reach the thresholds or were discarded by a third organism retaining both genes (*Zahn-Zabal, Dessimoz & Glover, 2020*).

## OrthoFinder

OrthoFinder defines an orthogroup as the set of genes derived from a single gene in the last common ancestor of the all species under consideration (*Emms & Kelly, 2015*). First, OrthoFinder performs all-*vs*-all blastp (*Camacho et al., 2009*) comparisons and uses an e-value of $1 \times 10^{-3}$ as a threshold. Then, it normalizes the gene length and phylogenetic distance of the BLAST bit scores. It uses the lowest normalized value of the RBH for either gene in a gene pair as the threshold for their inclusion in an orthogroup. Finally, it weights the orthogroup graph with the normalized bit scores and clusters it using MCL. OrthoFinder outputs the orthogroups and orthology relationships, which can be many to many (co-orthology).

We ran OrthoFinder with all the proteomes for each of the taxonomic groups listed (Table 1). From the OrthoFinder outputs with the orthology relationships between every two species, we used those considering the query organism. From an orthogroup containing one or more orthology relationships, we identified the outparalogs as those genes belonging to the same orthogroup but not to the same orthology relationship. We identified the inparalogs for the query organisms as its genes belonging to the same orthogroup since they derived from a single ancestor gene.

## ProteinOrtho

ProteinOrtho is a graph-based tool that implements an extended version of the RBH heuristic and is intended for the identification of ortholog groups between many organisms (*Lechner et al., 2011*). First, from all-*vs*-all blast results, ProteinOrtho creates subnetworks using the RBH at the seed. Then, if the second best hit for each protein is almost as good as the RBH, it is added to the graph. The algorithm claims to recover false negatives and to avoid the inclusion of false positives (*Lechner et al., 2011*).

We ran ProteinOrtho pairwise since we needed to identify orthologs and paralogs between the query organisms and the other members of their taxonomic data, not those orthologs shared between all the organisms. ProteinOrtho run blast with the following parameters: an e-value cutoff of $1 \times 10^6$, minimal alignment coverage of 50% of the shortest sequence, and 25% of identity. Orthologs were the genes from different genomes that belonged to the same orthogroup, and inparalogs genes from the same genome that belong to the same orthogroup. We reran ProteinOrtho with a similarity value of 75% instead of 90%, to identify outparalogs as those interactions not identified in the first run.

## Non-synonymous to synonymous substitutions

To perform *dN/dS* estimates, we used the CODEML program from the PAML software suite (*Yang, 2007*). The DNA alignments were derived from the protein sequence alignments using an *ad hoc* program written in PERL. The same program ran pairwise comparisons using CODEML to produce Bayesian estimates of *dN/dS* (*Angelis, dos Reis & Yang, 2014*; *Anisimova, Bielawski & Yang, 2002*). The results were separated between ortholog and paralog pairs, and the density distributions were plotted using R (*R Core Team, 2020*). Statistical analyses were also performed with R.

## Codon adaptation index

To calculate the Codon Adaptation Index (CAI) (*Sharp & Li, 1987*), we used ribosomal proteins as representatives of highly expressed genes. To find ribosomal proteins we matched the COG ribosomal protein families described by *Yutin et al. (2012)* to the proteins in the genomes under analysis using RPSBLAST (part of NCBI's BLAST+ suite) (*Camacho et al., 2009*). RPSBLAST was run with soft-masking (-seg yes -soft_masking true), a Smith-Waterman final alignment (-use_sw_tback), and a maximum e-value threshold of $1 \times 10^{-3}$ (-evalue 1e−3). A minimum coverage of 60% of the COG domain model was required. To produce the codon usage tables of the ribosomal protein-coding genes, we used the program *cusp* from the EMBOSS software suite (*Rice, Longden & Bleasby, 2000*). These codon usage tables were then used to calculate the CAI for each protein-coding gene within the appropriate genome using the *cai* program also from the EMBOSS software suite (*Rice, Longden & Bleasby, 2000*).

# RESULTS AND DISCUSSION

While we have been working on this report, an article following the same basic idea, comparing *dN/dS* distributions between orthologs and paralogs, though focusing on vertebrates, was published (*David, Oaks & Halanych, 2020*). Their results were consistent with those described below.

## Reciprocal best hits showed lower *dN/dS* ratios than paralogs

These studies used Bayesian *dN/dS* estimates, because they are considered the most robust and accurate (*Anisimova, Bielawski & Yang, 2002*; *Angelis, dos Reis & Yang, 2014*). To compare the distribution of *dN/dS* values between orthologs and paralogs, we plotted *dN/dS* density distributions using violin plots (Fig. 1). These plots demonstrated evident differences, with orthologs showing lower *dN/dS* ratios than paralogs, thus indicating that orthologs have diverged in function less frequently than paralogs. In line with the noticeable differences, Wilcoxon rank tests showed that the differences were statistically significant, with probabilities much lower than $1 \times 10^{-9}$ (Table S1). Since most comparative genomics work is done using reciprocal best hits (RBH) as a working definition for orthology (*Wolf & Koonin, 2012*; *Galperin et al., 2019*), this result suggests that most research in comparative genomics has used the proteins/genes that most likely share their functions.

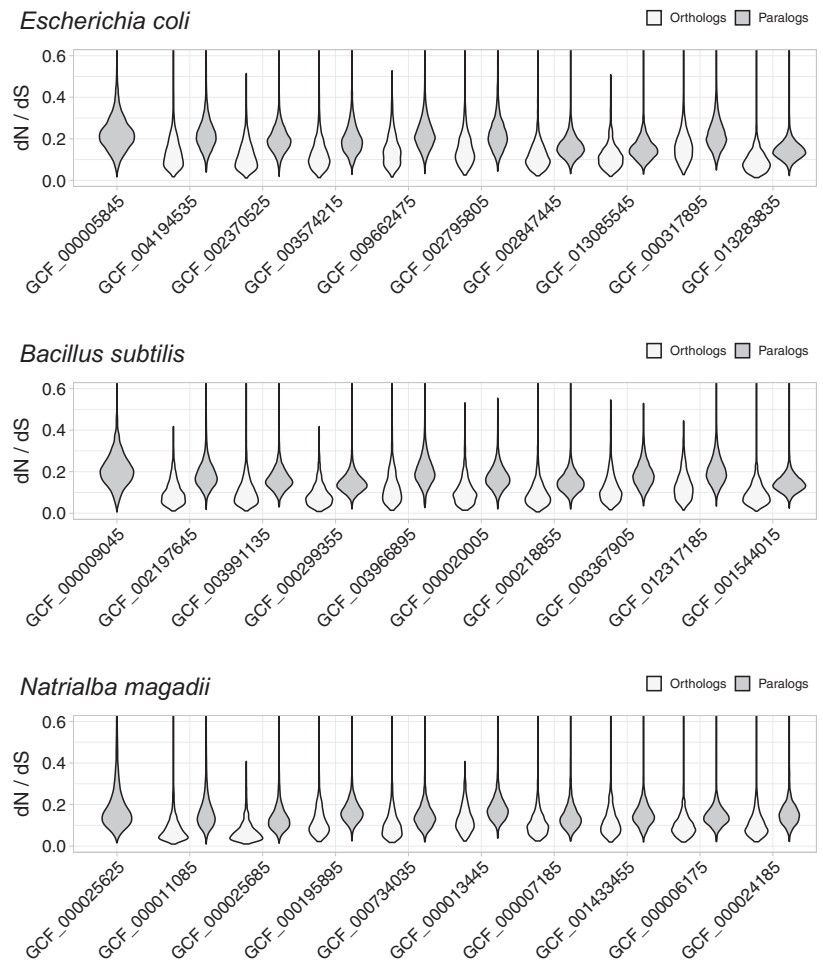

**Figure 1 Non-synonymous to synonymous substitutions (*dN/dS*).** The *dN/dS* ratios correspond to genes compared between query organisms against genomes from organisms in the same taxonomic phylum, namely: *E. coli* against other Proteobacteria, *B. subtilis* against other Firmicutes, and *N. magadii* against other Euryarchaeota. Genome identifiers are ordered from most similar to least similar to the query genome. The *dN/dS* distribution is higher for paralogs, suggesting that a higher proportion of orthologs have retained their functions.               

## Differences in *dN/dS* resisted other working definitions of orthology

A concern with our analyses might arise from our initial focus on reciprocal best hits (RBH). However, RBH might arguably be the most usual working definition of orthology (*Altenhoff & Dessimoz, 2009*; *Wolf & Koonin, 2012*; *Galperin et al., 2019*). Thus, it is important to start these analyses with RBH, at least to test whether RBH are a good choice for the purpose of inferring genes most likely to have similar functions.

Analyses of the quality of RBH for inferring orthology, based on synteny, showed that RBH error rates were lower than 5% (*Moreno-Hagelsieb & Latimer, 2008*; *Wolf & Koonin, 2012*; *Hernández-Salmerón & Moreno-Hagelsieb, 2020*). Other analyses showed that the problem with RBH, was a slightly higher rate of false positives (paralogs mistaken for orthologs), than databases based on phylogenetic and network analyses (*Altenhoff &*

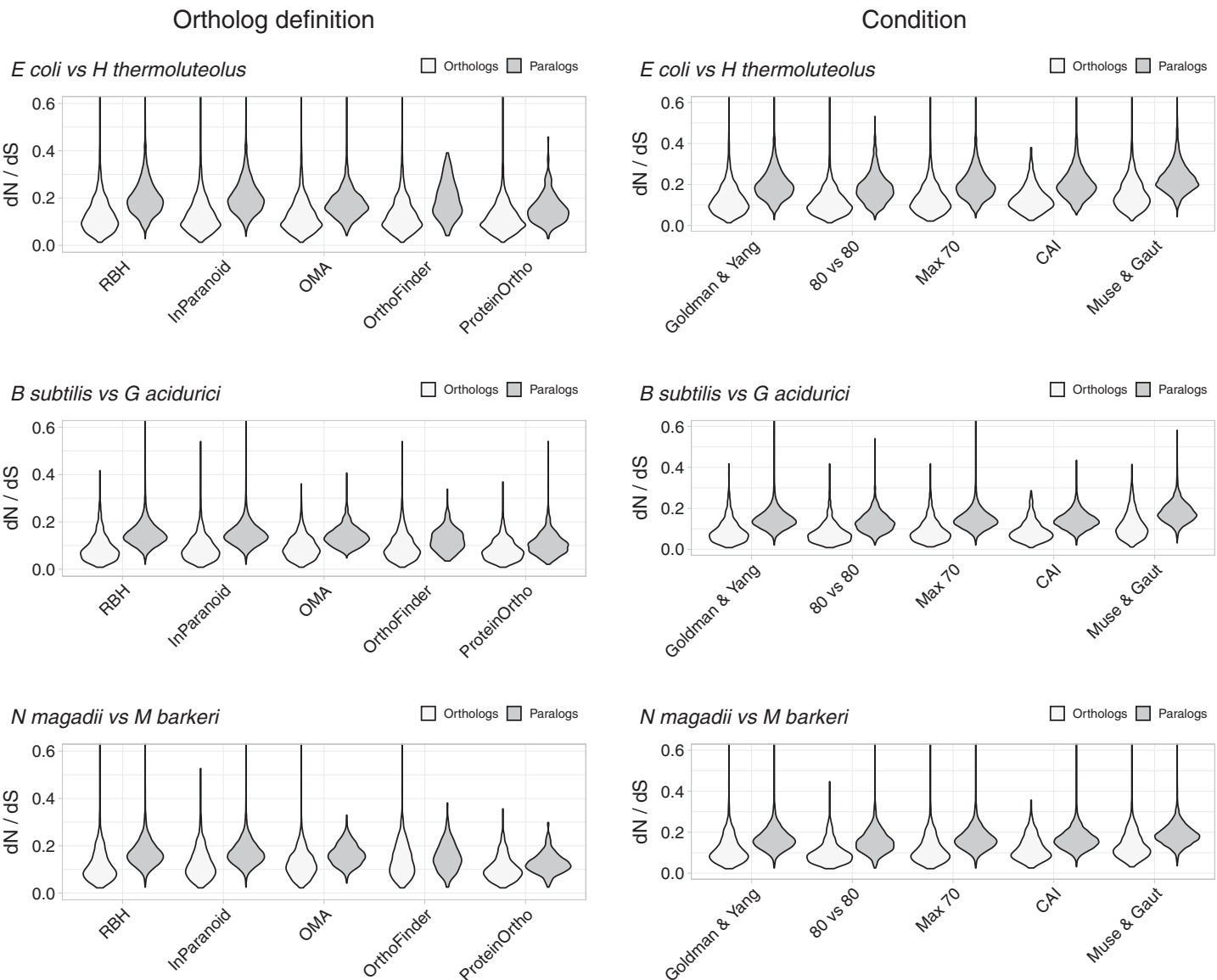

**Figure 2 Control experiments.** Left: values of *dN/dS* ratios were higher for different definitions of orthology than for their paralogs. RBH were included as reference. Right: examples of *dN/dS* values obtained testing for potential biases. The Goldman and Yang model for estimating codon frequencies (*Goldman & Yang, 1994*), included as reference, is the default. The 80 *vs* 80 test used data for orthologs and paralogs filtered to contain only alignments covering at least 80% of both proteins. The maximum identity test filtered out sequences more than 70% identical. The CAI test filtered out sequences having Codon Adaptation Indexes (CAI) from the top and bottom 15 percentile of the genome's CAI distribution. We also tested the effect of the Muse and Gaut model for estimating background codon frequencies (*Muse & Gaut, 1994*).

*Dessimoz, 2009*). Therefore, we can assume that orthologs dominate the RBH *dN/dS* distributions.

Despite the above justification for focusing on RBH, we considered four other definitions of orthology (Fig. 2, Figs. S1–S4). Orthologs obtained with different working definitions, including one method dealing with xenologs (OMA), showed *dN/dS* ratio distributions that suggest that a higher proportion of orthologs have similar functions compared to paralogs (Fig. 2).

## Differences in *dN*/*dS* persisted after testing for potential biases

While the tests above suggest that RBH separate homologs with higher tendencies to preserve their functions than other homologs, we tested for some potential biases. A potential problem could arise from comparing proteins of very different lengths. We thus filtered the *dN*/*dS* results to keep those where the pairwise alignments covered at least 80% of the length of both proteins. The results showed shorted tails in both density distributions, but the tendency for orthologs to have lower *dN*/*dS* values remained (Fig. 2, Fig. S5).

Another parameter that could bias the *dN*/*dS* results is high sequence similarity. In this case, the programs tend to produce high *dN*/*dS* ratios. While we should expect this issue to have a larger effect on orthologs, we still filtered both datasets, orthologs and paralogs, to contain proteins less than 70% identical. This filter had very little effect (Fig. 2, Fig. S6).

Lateral gene transfer events might be a problem with orthology predictions. However, proper genome-wide identification of lateral gene transfer events is difficult, as xenologs are hard to distinguish from duplications events (*Roth, Gonnet & Dessimoz, 2008*). Additionally, there is no good agreement between the output of different xenolog prediction methods benchmarked against real data (*Ravenhall et al., 2015*). In an attempt to deal with xenologs we used two approaches: We removed genes with atypical codon usage bias (see below), besides including an orthology working definition (OMA), that attempts to deal with xenologs. OMA uses a verification step to help reduce the number of xenologs by using a third proteome as witness of non-orthology (*Roth, Gonnet & Dessimoz, 2008*).

As mentioned above, to try and avoid the effect of sequences with unusual compositions, we filtered out sequences with extreme codon usages as measured using the Codon Adaptation Index (CAI) (*Sharp & Li, 1987*). For this test, we eliminated sequences with CAI values from the top and the bottom 15 percentile of the respective genome's CAI distribution. After filtering, orthologs still exhibited *dN*/*dS* values below those of paralogs (Fig. 2, Fig. S7).

Different models for background codon frequencies can also alter the *dN*/*dS* results (*Bielawski, 2013*). Thus, we performed the same tests using the Muse and Gaut model for estimating background codon frequencies (*Muse & Gaut, 1994*), as advised in (*Bielawski, 2013*). Again, the results showed orthologs to have lower *dN*/*dS* ratios than paralogs (Fig. 2, Fig. S8).

## Differences in *dN*/*dS* ratios were more evident for genes encoding for less divergent proteins

Orthologs will normally contain more similar proteins than paralogs. Thus, a similarity test alone would naturally make orthologs appear less divergent and, apparently, less likely to have evolved new functions. While synonymous substitutions attest for the strength of negative/purifying selection in *dN*/*dS* analyses, seemingly making these ratios independent of the similarity between proteins, we still wondered whether the data changed with protein sequence divergence.
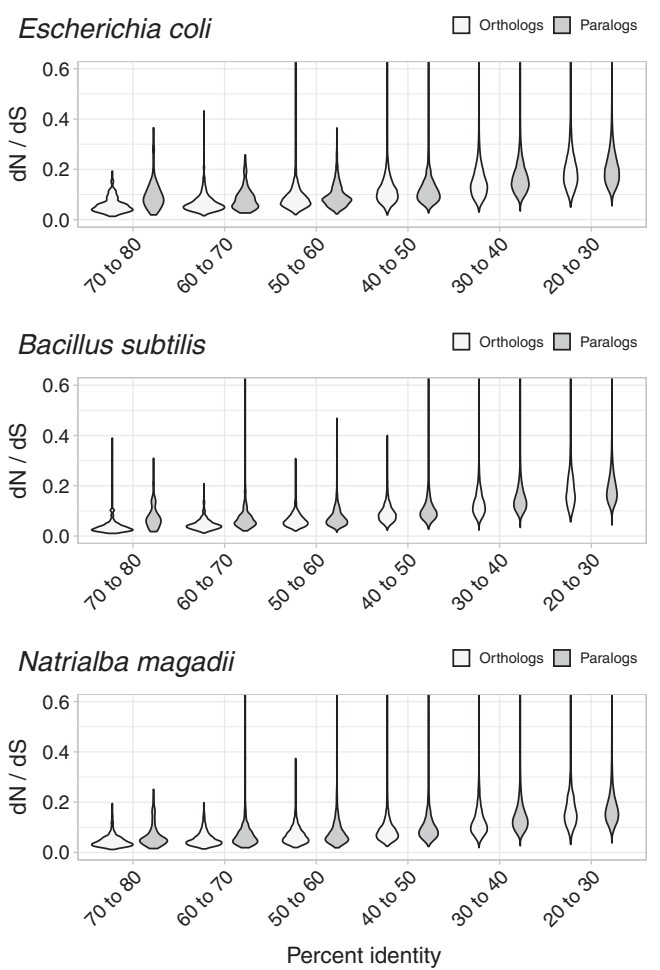

**Figure 3 Non-synonymous to synonymous substitutions *dN/dS* and divergence.** The difference between *dN/dS* ratios became less apparent as protein identity decreased.

To test whether *dN/dS* increased against sequence divergence, we separated orthologs and paralogs into ranges of divergence of the encoded protein's percent identity. The more similar the protein sequences, the more evident were the differences between the *dN/dS* of orthologs and paralogs (Fig. 3). Since protein sequence identity plays a role in most working definitions of orthology, the latter results partially explained the evident disparity in *dN/dS* ratios between orthologs and paralogs. However, that the ratio differences were more evident at low protein sequence divergence supports the hypothesis that paralogs might be an immediate source of functional novelty. Given that redundant duplications would be expected to eventually erode (*Ochman & Davalos, 2006*), early functional divergence might provide paralogs with the selective pressure to survive genetic erosion.

## CONCLUSION

The results shown above used a measure of divergence that relates to the tendencies of sequences to diverge in amino-acid composition, against their tendencies to remain unchanged; namely, non-synonymous to synonymous substitution rates (*dN/dS*). Since

changes in function require changes in amino-acids, this measure might suggest which sequence datasets have higher proportions that remain functionally coherent. Such proportions would show as a tendency towards lower $dN/dS$ values. Orthologs showed evidently lower values of $dN/dS$ than paralogs. Thus, orthologs could be though as more functionally stable than paralogs, with paralogs being a main source of novel functions.

## ACKNOWLEDGEMENTS

We are grateful to Joe Bielawski for helpful advice. We thank The Shared Hierarchical Academic Research Computing Network (SHARCNET) for computing facilities.

### Funding

Work supported with a Discovery Grant to Gabriel Moreno-Hagelsieb from the Natural Sciences and Engineering Research Council of Canada (NSERC). This work was also supported by the Programa de Apoyo a Proyectos de Investigacion e Innovacion Tecnologica (PAPIIT-UNAM) (IN205918 and IN202421) to Julio A. Freyre-González. Juan M. Escorcia-Rodríguez is supported by PhD fellowship 959406 from Consejo Nacional de Ciencia y Tecnología (CONACyT-Mexico). The funders had no role in study design, data collection and analysis, decision to publish, or preparation of the manuscript.

### Grant Disclosures

The following grant information was disclosed by the authors:
Natural Sciences and Engineering Research Council of Canada (NSERC).
Programa de Apoyo a Proyectos de Investigacion e Innovacion Tecnologica (PAPIIT-UNAM): IN205918 and IN202421.
PhD Fellowship 959406 from Consejo Nacional de Ciencia y Tecnología (CONACyT-Mexico).

### Competing Interests

The authors declare that they have no competing interests.

### Author Contributions

- Juan M. Escorcia-Rodríguez performed the experiments, analyzed the data, prepared figures and/or tables, and approved the final draft.
- Mario Esposito performed the experiments, analyzed the data, prepared figures and/or tables, and approved the final draft.
- Julio A. Freyre-González analyzed the data, authored or reviewed drafts of the article, and approved the final draft.
- Gabriel Moreno-Hagelsieb conceived and designed the experiments, performed the experiments, analyzed the data, prepared figures and/or tables, authored or reviewed drafts of the article, and approved the final draft.

## Data Availability

The programs for obtaining orthologs and dN/dS values as tested in this study are available at GitHub: https://github.com/Computational-conSequences/SequenceTools.

Genomes used in this study are available in Table 1.

## Supplemental Information

Supplemental information for this article can be found online at http://dx.doi.org/10.7717/peerj.13843#supplemental-information.

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
