# Peer review of "Non-synonymous to synonymous substitutions suggest that orthologs tend to keep their functions, while paralogs are a source of functional novelty"

_PeerJ, doi:10.7717/peerj.13843_

## Round 0.1 · original submission · Major Revisions

Dear Dr. Escorcia-Rodriguez and colleagues:

Thanks for submitting your manuscript to PeerJ. I have now received three independent reviews of your work, and as you will see, one reviewer recommended rejection, while another suggested a major revision. I am affording you the option of revising your manuscript according to all three reviews but understand that your resubmission may be sent to at least one new reviewer for a fresh assessment (unless the reviewer recommending rejection is willing to re-review).

The reviewers raised many concerns about the manuscript. Please address all of these in your rebuttal letter. In general, the reviewers wish to see improvements to English and grammar, as well as more careful treatments of orthology, paralogy and gene vs. protein function.

Good luck with your revision,

-joe

Reviewer 1 ·

Basic reporting

In this manuscript the authors look at the ratio of non-synonymous to synonymous substitutions following speciation and duplication events within several prokaryotic groups in order to test the expectation that ortholog function is more conserved than paralogs. The article is structured well, although grammatical and spelling errors did impede my understanding in some places.

Experimental design

The authors do a commendable job assembling a battery of various methods and filtering strategies to address potential problems. However, I do have several serious concerns. Firstly, none of the homology inference methods used by the authors consider xenologs (genes related through HGT). HGT is one of, if not the, primary sources of genetic novelty in prokaryotes, where xenologs can represent 80% of the genome (Lukjancenko et al. 2010). The authors make some attempt to deal with xenologs by removing genes with abnormal codon usage bias in some analyses, but I believe any study investigating homology in prokaryotes should explicitly consider the possibility of HGT in all analyses. Similarly, to my knowledge Orthofinder is the only method that explicitly infers orthology on the basis of its definition i.e. speciation events. As such, I believe Orthofinder should be the primary method used for this study with the rest included as supplemental. To do otherwise risks inviting circular logic, if orthologs are defined by their sequence similarity as is the case with RBH and other methods rather than their homologous relationships it should come as no surprise that selection between them is more conserved. On this point, how were paralogs determined in Fig 2A? should not each method have its own orthologs AND paralog distribution? Lastly, this study needs to report p-values for statistical tests between all comparisons in the main text and supplemental. It is also not clear to me why a ranked test was used, how were the data paired and why?

Validity of the findings

If the findings remain unchanged after experimental design concerns are considered I largely agree with the interpretation and conclusions of the authors. However, I do believe they overstate their findings in some cases. Notably, this study did not test functional differences between orthologs and paralogs and should avoid making definitive conclusions concerning the evolution of protein function, which may or may not directly correlate with dN/dS. Furthermore, even if function and dN/dS were the same, this study did not demonstrate that orthologs share function (as claimed on Line 162), only that orthologs maintain more similar functions over longer timescales than paralogs.

Additional comments

I would prefer the authors not use “keep their functions better” when describing evolutionary conservation as to me it implies orthologs “want” to keep their functions, and that paralogs are somehow not as successful. I would suggest the more neutral “maintain ancestral functions” (though be careful with making direct claims regarding function, as discussed above).
Minor in-line comments:
Line 19: It is not self-evident why orthologs would maintain function more than paralogs based on their origins alone, more context here is needed
Line 26: “readily” should be “ready”
Line 55: personal preference but I would avoid rhetorical questions in primary research articles
Line 56: “Nehrt et all (Nehrt et al., 2011)” should be “Nehrt et al. 2011”
Line 61: Note this work itself has methodological issues, see (Lukjancenko et al. 2010)
Line 63: “annotator bias” should be “annotation bias”
Line 113 (and elsewhere): “combinatorial” should be “combination” (I think?)

Reviewer 2 ·

Basic reporting

Well-written for a debate on orthology versus paralogy, conserved function versus neofunctionalization. This is an old debate with overwhelming publications on the subject.
Perhaps the originality with the present study is an analysis of prokaryotic genes, but this is a rather limited achievement considering that no clear comparative analysis between prokaryotes and eukaryotes is provided. When provided, the extensive literature references on the debate should be moved to discussion. Still, the rather limited self-contain of the present study will not be enough to validate the hypothesis (see below).

Experimental design

The experimental design is not suitable to validate the conclusion. Mainly the function of the genes investigated is unknown, which is a critical point to assess whether the function is conserved or not.

Validity of the findings

Novelty is limited to dN/dS in orthologous/paralogous genes in bacteria.
Not all domains from genome ID are assessed to test the hypothesis.
This is only a limited set of domains across the same classes of bacteria.
How could the function diverge in such a limited amount of classes and phenotypes?
Then a gene may duplicate without a change in function. The duplication rate may be as important as mutation sites to associate evolution of gene family and phenotype.

Additional comments

I even disagree with the very first sentence of this article, that “orthologs diverge after speciation events and paralogs after gene duplications”. Orthologous genes may have contributed to new phenotypes and therefore may diverge before speciation. Then, I also disagree with the second assumption “orthologs are expected to keep their function better, while paralogs could be a source of new function”. Gene may duplicate to reinforce the same function within the same species.

The description of the assumption is using references dating back to twenty years ago, so it is not a novel idea to explore, even though a recent article (David et al. 2020) already published the author’s opinion on the assumption (see line 72), just an old matter of discussion in the field of comparative genomics. In the discussion of the idea, it should be remembered the simple definition of orthologous (duplicates within two species) and paralogous (duplicates within the same species). Like mentioned by the authors (and others), whether orthology associates with conserved function is a huge debate, which needs to be tackled in the Discussion section, not in Introduction.

If the originality of the present study resides in the exploration of the idea in prokaryotes, it must be presented more accurately, mentioning differences between prokaryote and eukaryote genomes and their specific path of evolution, including RNA editing and protein diversity. Because what it is very important here is not much that using bioinformatics tools seems to be in favor of orthology and conserved function, it is to analyze a gene family for which function (or multifunction) is known. Then, if the gene family is conserved from bacteria to vertebrates, it will be lucky enough to have a suitable model to test the hypothesis that two orthologous genes have conserved functions. Without any data on the gene product, i.e. the protein, nothing can be said in term of function on the basis of dN/dS. Why? Because there are multiple posttranslational processes (which are not considered here) which could lead to a protein with a new function without gene duplication.

dN/dS is useful for an evolutionary study of a gene family, assessing which type of mutation is associated with a twist of function (and adaptation), but no conclusion can be drawn to infer neofunctionalization from gene duplication in this type of analysis. Protein data are required. What is the function of the bacterial proteins analyzed in this study by dN/dS software?

Reviewer 3 ·

Basic reporting

The manuscript investigates the "orthologs conjecture”—the notion that orthologs are functionally more conserved than paralogs—by comparing their synonymous versus non-synonymous substitution rates (dN/dS). The underlying assumption is that stronger purifying selection (indicated by a lower dN/dS) is indicative of functional conservation.

The paper is generally well written and clear. The style is at times a little informal, but it makes it easy to follow.

I advise being more careful in the introduction not to dismiss *from first principles alone* the possibility that paralogs could be functionally more conserved than orthologs:
- "it is very hard to think of a scenario whereby orthologs would diverge in functions at a higher rate than paralogs.". Here are potential scenarios for same-species paralogs: gene conversion, the notion that same-species paralogs, being in the same genomes, are potentially subject to the same evolutionary pressure.
- "Again, it is very hard to imagine a scenario where orthologs, as a group, would diverge in functions more than paralogs". Redundant.

One discretionary changes I would suggest is to remove the first sentence/paragraph of the introduction, which is redudant with the title and the abstract. "In this report, we present an analysis of non-synonymous and synonymous substitutions (dN/dS) what suggests that orthologs keep their functions better than paralogs."

Experimental design

The work is certainly within the aims of the scope of the journal, the research question is very well defined, is relevant, and meaningful. The work is well contextualised in terms of previous relevant work. The methods are described with sufficient details and information to replicate the work,

Validity of the findings

The authors report consistently lower dN/dS for orthologs, across a wide range of prokaryotes as well as the human-mouse pair. This result seems highly robust.

It would be customary to support the findings with statistical tests, but the difference in the plot is so clear that I view this point as a discretionary suggestion.

The authors have implemented several controls, including testing different methods for ortholog assignment, by filtering the data according to different criteria (alignment coverage, pairwise similarity, codon adaptation index, codon frequencies).

The conclusions are generally warranted, but again, I would caution against using first-principles: "It would also be proper to stop referring to the now confirmed expectation as a conjecture, since the expectation arises naturally from the definition.". I agree that it would be warranted to stop referring to this as a conjecture, but the reason for that is not that the expectation arises naturally from the definition, but rather that a growing body of evidence, including this work, has confirmed it.

---

## Round 0.2 · Minor Revisions

Dear Dr. Escorcia-Rodriguez and colleagues:

Thanks for resubmitting your manuscript. Two reviews are mostly positive and recommend acceptance; however, reviewer 2 has raised more concerns. Please look over these concerns raised by reviewer 2 and revise your work accordingly.

Good luck with your revision,

-joe

Reviewer 1 ·

Basic reporting

Overall the article remains well-written and more than meets the requirements of the journal. I still believe the structure of some sentences could be edited for clarity, or broken up into multiple parts to avoid run-ons. In particular, I had to parse the second half of the abstract multiple times. However this is a minor point that did not unduly obstruct my understanding of the manuscript on the whole.

Experimental design

The authors made considerable effort to address my main concern, which was the presence of xenology. The authors include a method which explicitly tests for xenology in addition to previous filtering steps, as well as adding a caveat outlining the potential problems associated with horizontal gene transfer events. My other concerns have been similarly addressed to my satisfaction.

Validity of the findings

The authors have made considerable changes to improve the robustness of their data and results, in particular by including plots and statistical tests across all relevant comparisons between orthologs and paralogs. With this framework in mind I consider their results and conclusions valid, and appreciate the changes they have made to the language to make their findings more grounded within the context and scope of their study.

Additional comments

I have several minor in-line comments. While I believe these suggestions would benefit the manuscript, they are not a requirement for my recommendation to accept.

Line 58: I would use a different example, these proteins in particular have been experimentally characterized extensively, and I doubt Orthology-based inferences play much of a role in our understanding of their functions.

Line 92: Since you directly mention your results in comparison with another study, I would consider moving this paragraph to the discussion.

Line 222: If none of the results from the Yang and Nielsen approximation method are presented in the current version of the manuscript, I do not think this sentence is needed.

Lastly, I find the frequent inclusion of the phrase "as a group" unnecessary when referring to orthologs/paralgos and believe it interrupts the flow of the manuscript. As a reader I understand the authors are not literally referring to every single ortholog and are instead discussing broad, general trends. However, I understand the authors may be trying to address the concerns of another reviewer, and are therefore unable to placate both of us.

Reviewer 2 ·

Basic reporting

Interesting topic: orthology maintains ancestral function, paralogy mediates innovation
Rather well-written, except conclusions
Deserves publication with some restrictions in the conclusion analysis and more background/literature on the protein function throughout the whole text
Two more figures (genomic organization and phylogenetic tree) should be required
Relevant results to hypothesis as long as the hypothesis strengthens the hypothesis, not a conclusion.

Experimental design

Material and Methods are limited to bioinformatic analysis.

Validity of the findings

Conclusions should be limited to bioinformatic analysis.

Additional comments

I still disagree with the very first sentence of this article, that “orthologs diverge after speciation events and paralogs after gene duplications”. The definition of “divergence” implies functional divergence. Divergence pertains to an evolutionary process wherein a population of an inbreeding species diverges into two or more descendant species that have become more and more dissimilar in terms of forms and structures. General divergence? I think the authors mean to say “separate”. Diverge means equal depart but usually suggests a branching of a main path into two or more leading in different directions. Orthologous genes may have contributed to new phenotypes and therefore may ‘diverge’ before speciation is important to note. Here, it is probably important to remind about definitions on common ancestry, evolution and duplication events. Gene duplications are an essential source of genetic novelty that can lead to evolutionary innovation. Duplication creates genetic redundancy, where the “second” copy of the gene is often free from selective pressure. The simple definition of orthologs (duplicates within two species) and paralogs (duplicates within the same species) refers to common ancestral gene and one gene copy (for innovation or reinforcement of the function). The notion of “clusters”, instead of “groups”, is also important when referring to evolution and diversification of functions. Then, the second assumption should also be rephrased “orthologs are expected to remain more functionally coherent across lineages, while paralogs have been proposed as a source of new functions” or deleted.

Keeping in mind the notion of ancestral gene or function, orthology/paralogy, gene family, gene cluster and duplication (eventually retained without acquiring new functions), the work presented here would deserve publication with major revisions. Major revisions start with the title which should probably be: “Bioinformatic dN/dS analysis on Bacterial and Archeal genes: orthologs tend to keep the ancestral function, paralogs make the source of functional innovation”. The model of Bacterial and Archeal genes should be described or explained in detail: which gene family is studied? How many genes? How many duplications within a species? How many gene clusters and intron/intronless genes? What is known about the function(s)? A figure showing the schematic organization of the genes would be a relevant addition. A figure showing the phylogenetic distribution of the genes would be even more relevant addition before the dN/dS analysis. Text should be rewritten with greater caution when it comes to the conclusion of the outcome from bioinformatics analysis. Discussion should not repeat Introduction too closely. Conclusions should be thoroughly revised –the entire section should be rewritten focusing on the model, bioinformatics, and main finding, to gain clarity. Sentences such as “It would also be proper to stop referring to the now confirmed expectations as a conjecture. Finally, we did not expect the differences to be so evident” should be removed. It should be said that “Based on bioinformatics analysis (dN/dS), our results suggest that paralogs tend to acquire novel functions, while orthologs retain the ancestral function, although this needs to be confirmed by analysis of the protein functions. It should be eventually mentioned that a possible perspective of this work would be to combine non-synonymous to synonymous substitutions to RNA editing, peptide editing and DNA/RNA dependent polymerization processes as those described in Xuan et al. (2014, 2016, 2019).
See Picimbon JF. A new view of genetic mutations. Australas Med J 2017; 10: 701-715.
See also Picimbon JF. Evolution of protein physical structures in insect chemosensory systems. In: Olfactory Concepts of Insect Control - Alternative to Insecticides; Picimbon, J.F., Ed.; Springer Nature: Switzerland AG, 2019, Vol. 2, pp. 231-263, as an interesting model of protein structure, mutation and function evolution in a cluster of paralogous genes.

Reviewer 3 ·

Basic reporting

The points I raised in my previous reports were satisfactorily addressed. I congratulate the authors for the careful additional work.

I just noticed one typo in this new version:
- l. 196 "other four" → "four other"

Experimental design

The points I raised in my previous reports were satisfactorily addressed.

Validity of the findings

The points I raised in my previous reports were satisfactorily addressed.

---

## Round 0.3 · accepted · Accept

Dear Dr. Escorcia-Rodriguez and colleagues:

Thanks for revising your manuscript based on the concerns that were raised. I now believe that your manuscript is suitable for publication. Congratulations! I look forward to seeing this work in print, and I anticipate it being an important resource for groups studying molecular evolution. Thanks again for choosing PeerJ to publish such important work.

Best,

-joe